# Exploring Definitions of “Addiction” in Adolescents and Young Adults and Correlation with Substance Use Behaviors

**DOI:** 10.3390/ijerph19138075

**Published:** 2022-06-30

**Authors:** S. Elisha LePine, Elias M. Klemperer, Julia C. West, Catherine Peasley-Miklus, Caitlin McCluskey, Amanda Jones, Maria Roemhildt, Megan Trutor, Rhonda Williams, Andrea Villanti

**Affiliations:** 1Vermont Center on Behavior and Health, Department of Psychiatry, University of Vermont, Burlington, VT 05405, USA; elias.klemperer@med.uvm.edu (E.M.K.); julia.west@uvm.edu (J.C.W.); catherine.peasley-miklus@med.uvm.edu (C.P.-M.); caitlin.mccluskey@uvm.edu (C.M.); andrea.villanti@rutgers.edu (A.V.); 2Department of Psychology, University of Florida, Gainesville, FL 32611, USA; 3Department of Psychological Science, University of Vermont, Burlington, VT 05405, USA; 4Health Surveillance, Vermont Department of Health, Burlington, VT 05402, USA; amanda.jones@vermont.gov (A.J.); maria.roemhildt@vermont.gov (M.R.); 5Alcohol & Drug Abuse Programs, Vermont Department of Health, Burlington, VT 05402, USA; megan.trutor@vermont.gov; 6Health Promotion & Disease Prevention, Vermont Department of Health, Burlington, VT 05402, USA; rhonda.williams@vermont.gov; 7Rutgers Center for Tobacco Studies, Department of Health Behavior, New Brunswick, NJ 08901, USA

**Keywords:** youth, young adult, addiction, substance use, intervention, messaging, mixed methods, qualitative

## Abstract

(1) Background: Young people engage in addictive behaviors, but little is known about how they understand addiction. The present study examined how young people describe addiction in their own words and correlations between their definitions and substance use behaviors. (2) Methods: Young adults (*n* = 1146) in the PACE Vermont Study responded to an open-ended item “what does “addiction” mean?” in 2019. Responses were coded using three inductive categories and fifteen subcategories. Quantitative analyses examined correlations between addiction theme definitions, demographics, and substance use behaviors. (3) Participants frequently defined addiction by *physiological* (68%) and *psychological* *changes* (65%) and less by *behavioral changes* (6%), or all three (3%); young adults had higher odds of defining addiction as *physiological* or *behavioral* *changes* than adolescents. Participants who described addiction as “*psychological changes*” had lower odds of ever electronic vapor product use (OR 0.75, 95% CI 0.57–1.00) than those using another definition, controlling for age and sex. (4) Perceptions of addiction in our sample aligned with existing validated measures of addiction. Findings discriminated between familiar features of addiction and features that may be overlooked by young adults. Substance users may employ definitions that exclude the symptoms they are most likely to experience.

## 1. Introduction

Adolescence (12–17 years old) and young adulthood (18–25 years old) are developmental stages associated with vulnerability to addiction and high rates of risky behaviors [1,2,3,4]. Substance use initiated during these periods can persist into older adulthood [5] and develop into long-term chronic health problems later in life [6,7]. Communicating the risks associated with substance use early in life is important to public health efforts to prevent long-term health harm.

Mass reach media communication and education campaigns have played a key role in substance use prevention efforts over the last 50 years, to varying degrees of success [8,9,10,11,12,13]. Effective mass media public education campaigns on tobacco have targeted beliefs as the precursor to changing attitudes and, ultimately, tobacco use behavior [10,11,14,15,16,17]. These pathways have been empirically supported by evaluations of the Truth campaign [16,17] and serve as the basis for the U.S. Food and Drug Administration’s (FDA) Real Cost campaign [14,15,18,19]. 

In designing such media campaigns, the selection of maximally effective messaging themes is crucial. Brennan and colleagues [20] found that messaging about addiction was among the top three potentially effective themes in federally funded mass media prevention campaigns targeting cigarette smoking in youth. The FDA’s The Real Cost campaign has also targeted the theme of loss of control due to addiction in its smoking prevention messaging [21,22]. More recently, addiction has been identified as a key theme for e-cigarette prevention messaging [23], including iterations of the FDA’s Real Cost campaign [21]. Given the relationship between early initiation of cannabis use and cannabis dependence [24], addiction may be a salient theme for cannabis prevention messaging, though research on such messaging is currently lacking [25,26]. 

Little is known about the specific addiction-related beliefs held by youth and young adults or the most effective ways mass media campaigns can target these beliefs. In a series of focus groups for the development of messages for The Real Cost campaigns, Roditis and colleagues [27] found that youth were most receptive to addiction-themed messages emphasizing the immediate, concrete consequences of addiction, as well as its social consequences. However, some reported the consequences of addiction in these messages were overblown [27]—a finding consistent with research showing that young people may fail to recognize if and when they have an addiction and underestimate the risk of developing an addiction [28,29]. In contrast, a recent qualitative study of teens’ and young adults’ motivations for quitting e-cigarettes found “freedom from addiction” among the top reasons for making a quit attempt [30]. The present study aimed to understand how young people describe addiction in their own words and how this may relate to substance use experience to support future prevention efforts targeting addiction.

## 2. Materials and Methods

Data for the current analyses come from Waves 1 and 2 of the Policy and Communication Evaluation (PACE) Vermont Study conducted on Vermont adolescents (aged 12–17) and young adults (aged 18–25) in Spring and Summer 2019. Participants were recruited via paid and unpaid online advertising, community recruitment through affiliated organizations, and participant-to-peer referrals [31]. Eligible participants took three online surveys from March 2019 through October 2019. These surveys contained measures of past and current substance use, beliefs about substances, opinions on state and national policies related to substances, and awareness of mass media education campaigns. This research was approved by the University of Vermont’s Institutional Review Board and received a Certificate of Confidentiality from the National Institutes of Health. Detailed study methods have been described elsewhere [31]. Of the 1497 participants who completed the Wave 1 survey, 1166 (78%) also completed the Wave 2 survey. The final analytic sample included 1146 participants who received the open-ended item on addiction and were matched back to the full dataset (385 youth and 761 young adults).

### 2.1. Measures

At Wave 2 (Summer 2019), participants received an open-ended response item that captured participants’ understandings of addiction in their own words, asking “What does ‘addiction’ mean?” The item offered no additional instruction to avoid priming participants. Participants entered their responses in a single-line field that appeared on screen, with no minimum or maximum character limit. 

Participant demographics were collected during Wave 1 (Spring 2019) of the PACE Vermont Study and included age group, sex, and race/ethnicity. Age group was coded as youth (ages 12–17) or young adult (ages 18–25), sex was coded as male or female, and race/ethnicity was coded as white versus non-white. Wave 2 items also captured ever and past 30-day use of cigarettes, electronic vapor products (EVPs), alcohol, and marijuana among all participants. Ever and past 30-day substance use was defined as endorsement of use of any of the substances assessed (i.e., cigarettes, EVPs, alcohol, marijuana).

### 2.2. Qualitative Analyses

The entire body of responses to the open-ended question “what does ‘addiction’ mean?” was reviewed by one member of the research team (SEL); 36 participants were missing a response to this item and were excluded from subsequent review, yielding 1110 responses to be coded. Common ideas and phrases were noted, and an initial coding manual was developed from these. This coding manual was discussed with two additional members of the research team (JCW and ACV) and was edited for clarity and to reduce redundancy. Two coders (SEL and CM) tested the coding manual on a random set of 200 responses to ensure the usability of the codebook and acceptable agreement between coders. Following this initial training period, the coders met to resolve any disagreements and clarify the definitions of codes. The final coding manual contained 15 coding categories. The full list of coding categories, their definitions, and examples of responses coded into each category can be found in Table 1. The full body of responses was then coded independently by both coders using NVivo 12 software [32]. Both coders resolved disagreements by discussing each case of disagreement until a consensus was reached. Finally, the codes were assessed for major connecting themes that could capture the general types of responses. Both coders agreed to the use of *physiological changes*, *psychological changes*, and *behavioral changes* as thematic categories. Thematic categories highlight key differences in how respondents characterized addiction – specifically, the domain in which participants saw it operationalized. In addition to thematic categories, an additional code (N/A) was used to designate responses that were uninterpretable, off topic, or otherwise un-codable. All thematic categories and subcategories except N/A and its categories subcategories were non-exclusive; responses could be coded in multiple categories. Reliability for the codebook was assessed with Cohen’s kappa, which ranged from κ = 0.46 to κ = 0.89 for the thematic categories and subcategories, representing moderate to near-perfect agreement between coders. Response frequencies in each thematic category and subcategory were compared to identify the themes most and least salient for our sample.

### 2.3. Quantitative Analyses

Multivariable logistic regression models examined correlations between the three primary thematic coding categories and demographic characteristics, using the coding categories as the exposure variables and the demographic characteristic as the outcome of interest. Where there were significant correlations between a thematic category and a demographic characteristic, an additional model examined the relationship between that theme’s subcodes and the demographic characteristic (e.g., if *psychological changes* was significantly correlated with age group, a separate multivariable model examined the correlation between *psychological changes* subcategories and age group). A similar two-stage approach was used to examine correlations between addiction themes, ever, and past 30-day use of tobacco, electronic vapor products, marijuana, alcohol, and any substance. Given positive correlations between addiction themes, age, and sex, all multivariable models for substance use controlled for age and sex. Qualitative responses regarding tolerance (physiological changes) were recategorized into dependence for quantitative analyses (*n* = 3).

## 3. Results

### 3.1. Demographics

Of the 1146 participants in the analytic sample, 97.0% (*n* = 1110) responded to the open-ended addiction question. Adolescents made up 33.5% (*n* = 372) of respondents and young adults made up 66.5% (*n* = 738). The mean age of the sample was 19.1 years. Slightly more than one-quarter (28.0%; *n* = 311) of respondents were male, 71.8% (*n* = 797) were female, and 0.2% (*n* = 2) were missing data on sex. The majority of respondents (88.0%, *n* = 977) identified their race or ethnicity as white, 7.4% (*n* = 82) as non-white or other race, and 4.5% (*n* = 50) as Hispanic; one respondent was missing data on race/ethnicity. See Appendix A for a description of sociodemographic characteristics by adolescent versus young adult age groups. Most participants reported ever use of cigarettes, electronic vapor products (EVPs), alcohol or marijuana (*n* = 850, 76.6%). The rate was higher for young adults (*n* = 706, 96%) than for youth (*n* = 144, 39%).A majority of participants (*n* = 685, 61.7%) also reported past 30-day use of one or more of these four substances at Wave 2. Again, this rate was much higher for young adults (*n* = 609, 83%) than for youth (*n* = 76, 20%). Compared with adolescents, young adults were more likely to be female (77% vs. 63%; *p* < 0.001) and reported a higher prevalence of ever substance use (96% vs. 39%; *p* < 0.001) and any past 30-day substance use (83% vs. 20%; *p* < 0.001; Appendix A).

### 3.2. Frequencies of Coding Categories and Themes

Three thematic addiction coding categories emerged: *physiological changes*, *psychological changes*, and *behavioral changes*. These three categories captured the majority of responses (*n* = 1065, 96%), with only 4% (*n* = 45) being exclusively coded within *N/A* and its subcategories (Figure 1). Definitions most frequently referenced *physiological* changes, with 763 (68.7% of all responses) responses coded in this thematic category (Table 1) and 322/763 (42.2%) exclusively coded here. Responses in this category included “your body craves it and might have withdrawn symptoms if it does not receive it” (female young adult). Within the *physiological changes* thematic category, *physiological dependence* was the most prevalent subcategory (*n* = 645/763, 84.5%). This subcategory also contained two further subordinate categories: *functioning* (*n* = 231/645, 35.8%) and *withdrawal* (*n* = 92/645, 14.3%), though most responses (*n* = 322/645, 49.9%) in the *dependence* subcategory defined addiction as a physiological dependence with no greater level of specificity (e.g., “The body’s chemical dependence on a substance”—female young adult). Other subcategories within the physiological changes thematic category were *cravings* (*n* = 127/763, 16.5%) and *negative health consequences* (*n* = 54/763, 7.1%). The least-referenced subcategory within physiological changes was *organic disease* (i.e., explicitly defining addiction as a disease of brain or body, e.g., “it’s a disease”—female young adult), which accounted for 38 responses (5% of responses within the *physiological* category). *Tolerance* was identified as a subcategory in physiological changes but was mentioned only three times by study participants.

*Psychological changes* was the second thematic coding category, containing 719 (64.8% of total responses) responses, with 281/719 (39.1%) exclusive to this thematic category. Responses in this thematic category included “mentally you may need substances to feel like you can cope with negative emotions” (female young adult). Within *psychological changes*, *psychological need* (*n* = 370/719, 51.5%) *and self-regulation* (*n* = 371/719, 51.6%) were the most prevalent subcategories. *Affect* was another subcategory with 14.3% (*n* = 103/719) of responses. The least referenced subcategory within psychological changes was *cognition* (i.e., referencing changes to thought patterns, including planning around addiction or constantly thinking about it), accounting for 54/719 responses (7.5%). *Mental illness* was identified as a subcategory in psychological changes and was mentioned four times in these definitions. 

*Behavioral changes* were the third thematic coding category, containing 66 responses (6.0% of all responses), eight (12.1%) of which were exclusive to this thematic category. Responses in this thematic category included “When you go out of your way to make sure you always have a supply of it” (female young adult). Within this thematic category, responses primarily concerned addiction’s impact on functioning (*impaired functioning*), which accounted for 43 of the 66 responses in this category (65.2%). *Seeking behavior* (*n* = 20/66, 30.3%) and *relationship changes* (*n* = 15/66, 22/7%) were other subcategories within behavioral changes.

Overlap between psychological and physiological definitions was common, with 396 responses (36% of all responses) shared between them (Figure 1). Few respondents provided a definition that was coded at least once into each of the three primary categories and could be described as a holistic understanding of addiction (*n* = 29/1110 responses; 2.6%). 

### 3.3. Correlations between Thematic Categories and Demographics

Primary thematic category differed by age group, with greater endorsement of *physiological* (OR 1.61, 95% CI 1.22–2.12) or *behavioral changes* (OR 3.00, 95% CI 1.51–5.97) among young adults compared with adolescents (Table 2). Young adults also had higher odds of endorsing the *dependence* subcode under *physiological changes* (OR 1.69, 95% CI 1.30–2.18) than adolescents. Thematic category also differed by sex, with a lower likelihood of endorsing *psychological* (OR 0.69, 95% CI 0.52–0.92) or *physiological changes* (OR 0.71, 95% CI 0.53–0.96) among males than females. Males were also less likely to endorse the *psychological* subcode need (OR 0.72, 95% CI 0.54–0.97). Responses did not differ when comparing white to non-white respondents. 

### 3.4. Relationships between Thematic Categories and Substance Use

Where there were relationships between addiction themes and substance use in bivariate analyses, they were largely attenuated after controlling for age and sex in the multivariable models. There were no relationships between thematic categories and ever use of any substance, ever cigarette use, ever alcohol use, or ever marijuana use (Table 3), controlling for age and sex. The odds of endorsing psychological changes in the definition of addiction were lower for ever EVP users (OR 0.75, 95% CI 0.57–1.00) than never users (Table 3) in multivariable analyses; when subcodes for psychological changes were examined, this was largely driven by the lower odds of identifying self-regulation (e.g., “Loss of control when it comes to a behavior”) as a feature of addiction among ever users compared to never users of these products (OR 0.66, 95% CI 0.50–0.87). These analyses were repeated separately for youth and young adults. Among young adults, the odds of endorsing behavioral changes were lower for those who had ever used any substance (OR 0.57, 95% CI 0.33–0.98), and this relationship was driven by lower odds of endorsing the seeking behavior subcode (OR 0.22, 95% CI 0.08–0.64). Among youth, the odds of endorsing physiological changes were greater for those who had ever tried marijuana (OR 1.90, 95% CI 1.03–3.52), and this was driven by greater odds of endorsing the physiological dependence (OR 2.19, 95% CI 1.25–3.83) and functioning subcodes (OR 2.28, 95% CI 1.22–4.25).

## 4. Discussion

Three primary themes emerged from adolescent and young adults’ definitions of addiction: (1) *physiological* changes, (2) *psychological changes*, and (3) *behavioral changes*. These coding categories align with 8 out of 10 of the *Diagnostic and Statistical Manual of Mental Disorder’s* fifth edition (DSM-5) diagnostic criteria for substance use disorder [33], as well as existing parameters for assessing drug harms [34], demonstrating the consistency of the coding structure and our participants’ understandings of addiction with research on the defining features of addiction.

Comparisons between categories revealed the features of addiction that are salient to young people. The popularity of *physiological* and *psychological* responses, as well as the high degree of overlap between these categories, suggests that young adults perceive addiction largely as a matter of physiological dependence and a failure of psychological self-control. These responses could reflect a tension between involuntary physiological realities and personal choice. While many participants referenced the concept of dependence, these definitions were frequently circular or lacked further detail (e.g., “Being dependent on a substance”—female young adult). It is unclear whether this may be attributed to a vague understanding of the specific consequences of dependence on a substance or to the nature of the open-ended item. The prevalence of *self-regulation* aligns with past messaging campaigns targeting addiction, especially The Real Cost campaign, which effectively spoke to the loss of control caused by addiction to tobacco [22]. Messaging campaigns that provide concrete examples of how addiction constrains personal choice may be especially effective for these age groups, given adolescence and young adulthood are developmental stages marked by personal choice and identity exploration [2]. Messages that convey real-world consequences of dependence (e.g., altering routines to minimize craving or withdrawal) may supplement youth and young adults’ existing beliefs about the experience of addiction.

This study also identified important features of addiction that were not nominated by young people: (1) Our sample infrequently identified addiction as being either an organic disease or mental illness, but its appearance is notable as it speaks to awareness of some professional and clinical definitions of addiction. (2) Very few respondents mentioned tolerance, which is one of the diagnostic criteria for substance use disorder listed in the *DSM-5* [33]. (3) Respondents infrequently attended to behavioral changes associated with addiction, such as changes in social behavior, despite the inclusion of relationship changes as a result of addiction in existing clinical measures and guidelines (e.g., DSM-V) [34,35]. (4) While many participants defined addiction as relying on a substance to maintain basic functioning, far fewer referenced impaired daily functioning. The relative infrequency of responses coded as *behavioral changes* or to all three categories (holistic descriptions) indicates that education could produce a greater understanding of addiction in young people.

Differences in the themes by participant characteristics and substance use behaviors revealed distinct patterns between groups. For example, participants endorsing *psychological* and *physiological changes* had lower odds of being male than female, and this relationship held for those endorsing the *need* subcode. While there were few associations between addiction themes and ever substance use or past 30-day substance use, the *psychological changes* category was negatively correlated with ever EVP use, and this was largely driven by lower odds of EVP use among those who defined addiction in terms of lacking *self-regulation.* One explanation is that adolescents and young adults may define addiction separately from their own experiences—that is, they are motivated to think about addiction as being different from the consequences they have experienced while using substances. One possibility is that an individual currently using alcohol, for example, may be motivated to define addiction as a more severe physiological dependence and severely inhibited behavioral functioning than their current experience and less motivated to define addiction in terms of psychological features they face (e.g., having a hard time regulating how much they drink). However, a lack of data on the frequency of use limits our ability to confirm this specific hypothesis. Future research should therefore explore the role of motivated cognition and cognitive dissonance in adolescent and young adults’ perceptions of addiction, as well as how the full range of risks associated with addiction can be effectively communicated to this population.

## 5. Conclusions

To our knowledge, this study is the first to employ an inductive approach to understanding young people’s beliefs about addiction and the relationships between beliefs about addiction and substance use behavior. Findings from this study provide novel and valuable information to improve mass media communication campaigns that target youth and young adult beliefs about addiction. Results are limited in their generalizability by the geographic restriction to one state (Vermont), the use of a convenience sample, and larger proportions of female and young adult respondents in the sample. Beliefs about addiction were assessed with a single open-ended item, which may have resulted in greater numbers of vague, non-specific, or *I don’t know* responses than are representative of the sample’s beliefs about addiction. Future research should investigate adolescent and young adults’ beliefs about addiction in a national sample, as well as differences in beliefs by age, past and current substance use, and over time. 

Studies suggest that tobacco prevention messaging on health harms may be more effective than addiction messages [36,37,38]. However, prior research has indicated that messaging on the long-term threats to health is insufficient to overcome adolescents’ optimistic bias and prevent tobacco use [29]. Results from the current study suggest that substance use prevention messages addressing physiological dependence and cravings or loss of autonomy and self-control furnish immediate, concrete consequences that may resonate with young people [27]. Additionally, targeting less-familiar themes such as impacts on relationships and behavior change may broaden young people’s understanding of addiction.

## Figures and Tables

**Figure 1 ijerph-19-08075-f001:**
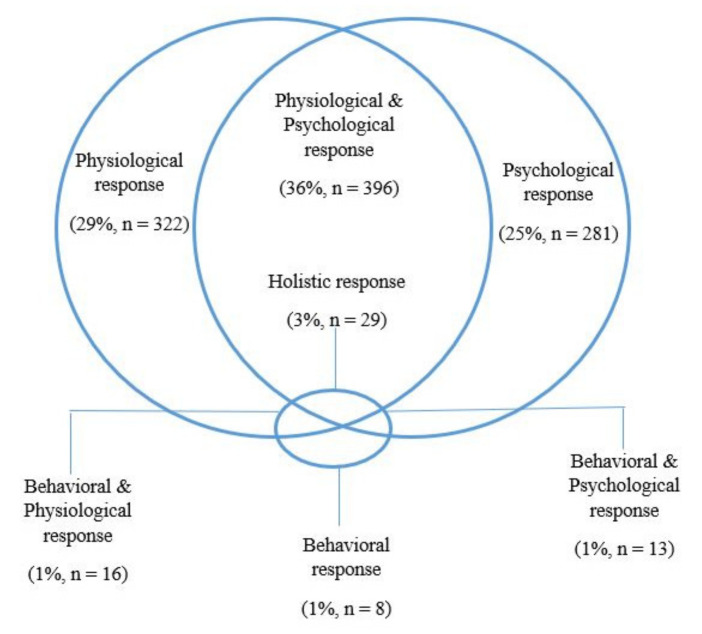
Overlap of responses coded in each category. Participants not coded into any major category (*n* = 45; 4%).

**Table 1 ijerph-19-08075-t001:** Addiction theme coding categories with definitions and examples, frequencies, and reliability.

Category/Subcategory ^a^ *Definition*			Example	Frequency % (n)	Reliability Cohen’s κ
**Physiological Changes** ** *Participant defines addiction with an emphasis on the physiological effects of addiction, including withdrawal and dependence.* **				68.7% (763/1110)	κ=0.79
	Physiological Dependence *Definition characterizes physiological dependence on a substance as a key component of addiction.*			84.5% (645/763)	κ = 0.72
		Functioning *Definition mentions relying on a substance to maintain basic functioning as a key component of dependence.*	*“Needing to do something in order to function properly” (male youth)*	35.8% (231/645)	κ=0.84
		Withdrawal *Definition mentions withdrawal symptoms as a key component of dependence*	*“Something you can’t stop doing without going into withdrawal.” (female youth)*	14.3% (92/645)	κ=0.83
	Cravings *Definition characterizes cravings as a key component of addiction (specifically those cravings experienced bodily, beyond merely “wanting” a substance).*		“*Addiction is an extreme craving.” (female youth)*	16.5% (127/763)	κ = 0.82
	Negative health consequences *Definition emphasizes the negative health consequences of addiction.*		*“Continuation of a behaviour* [sic] *despite one knowing it is harmful to themselves” (male young adult)*	7.1% (54/763)	κ = 0.46
	Organic disease *Definition characterizes addiction as a physiological condition or disease (including any mention of addiction as a disorder or as changing brain functioning or structure).*		*“Disease caused by chemical imbalance in the brain” (female young adult)*	5.0% (38/763)	κ = 0.80
**Psychological Changes** ** *Participant defines addiction with an emphasis on the motivational, cognitive, and/or affective changes resulting from addiction.* **				**64.8% (719/1110)**	κ=0.80
	Psychological need *Definition emphasizes the experience of “needing” the substance without reference to physiological cravings or dependence.*		*“…you keep using it until you feel you need it, or have to have it.” (female youth)*	51.5% (370/719)	κ = 0.63
	Self-regulation *Definition emphasizes failures of self-control leading to or caused by addiction.*		*“Loss of control when it comes to a behavior” (female young adult)*	51.6% (371/719)	κ = 0.78
	Affect *Definition emphasizes affective consequences of addiction (including mentions of distress when substance is inaccessible without a physiological component mentioned).*		“*You feel like you have to have more to be happy, feel normal, be calm or something like that.” (female young adult)*	14.3% (103/719)	κ = 0.71
	Cognition *Definition emphasizes the cognitive effects of addiction (e.g., changes in thought patterns, planning around substance use)*		*“People with a substance use disorder have distorted thinking”* *(female young adult)*	7.5% (54/719)	κ = 0.77
**Behavioral Changes** ** *Participant defines addiction with an emphasis on the behavioral changes in the addicted person without speculation as to (1) their psychological state or (2) their physiological state.* **				**6.0% (66/1110)**	κ=0.74
	Functioning *Definition emphasizes disruption of normal functioning as a key component of addiction (distinct from needing a substance in order to function normally).*		*“…functional impairment due to dependence.” (female young adult)*	65.2% (43/66)	κ = 0.70
	Seeking behavior *Definition mentions going out of one’s way to obtain the substance as a key component of addiction.*		*“Going out of your way to obtain and use a substance every day.if you don’t [sic] have it, it’s on your mind.” (female young adult)*	30.3% (20/66)	κ = 0.74
	Relationship changes *Definition emphasizes changes in relationships with others as a key component of addiction.*		*“when you can’t stop even though you know what it’s doing to you and the people close to you” (female young adult)*	22.7% (15/66)	κ = 0.52
**N/A** ***Comment not otherwise coded into content-based categories***.				**8.2% (91/1110)**	κ=0.47
	Other *Comment not able to be coded in other categories.*		*“So many things.” (female young adult)* *“A variety of things to different people” (female young adult)*	90.1% (82/91)	κ = 0.46
	Does not know *Participant expresses confusion or an inability to answer the question*		*“not sure” (female young adult)*	9.9% (9/91)	κ = 1.00

^a^ Categories not included in the table due to small cell sizes: Physiological–Tolerance (*n* = 3; <1%); Psychological–Mental illness (*n* = 3; <1%).

**Table 2 ijerph-19-08075-t002:** Logistic regressions of addiction themes by demographic characteristics, PACE Vermont Study 2019.

	*Age Group* *(Young Adults = 1;* *Adolescents = 0) * *(n = 1110)*	*Sex* *(Male = 1; * *Female = 0)* *(n = 1108)*	*Race/Ethnicity* *(Non-White or Hispanic = 1; * *White, non-Hispanic = 0)* *(n = 1109)*
	OR (95% CI)	OR (95% CI)	OR (95% CI)
** *Major themes* ** * ^a^ *			
*Psychological changes*	0.81 (0.61–1.07)	0.69 (0.52–0.92) **	0.90 (0.60–1.34)
*Physiological changes*	1.61 (1.22–2.12) ***	0.71 (0.53–0.96) **	0.74 (0.50–1.11)
*Behavioral changes*	3.00 (1.51–5.97) ***	0.89 (0.50–1.58)	0.59 (0.23–1.49)
** *Psychological changes ^b^* **		(n = 1108)	
*Affect*		0.96 (0.60–1.54)	
*Cognition*		0.57 (0.28–1.16)	
*Need*		0.72 (0.54–0.97) **	
*Self-regulation*		0.87 (0.66–1.16)	
** *Physiological changes ^b^* **	(n = 1110)	(n = 1108)	
*Craving*	1.09 (0.73–1.62)	0.69 (0.44–1.07)	
*Dependence*	1.69 (1.30–2.18) ***	0.80 (0.61–1.04)	
*Negative health consequences*	0.59 (0.34–1.03)	0.98 (0.53–1.81)	
*Organic disease*	1.50 (0.72–3.15)	0.78 (0.36–1.67)	
** *Behavioral Changes ^b^* **	(n = 1110)		
*Functioning*	3.11 (1.26–7.67) **		
*Relationship changes*	1.10 (0.28–4.27)		
*Seeking behavior*	2.75 (0.79–9.54)		

** *p* < 0.05; *** *p* < 0.001. ^a^ Multivariable logistic regression models include all three major themes in the same model. ^b^ Multivariable logistic regression models for subcodes were conducted separately (e.g., *psychological changes* model included only subcodes for *psychological changes* major theme).

**Table 3 ijerph-19-08075-t003:** Relationships between addiction themes and ever substance use, PACE Vermont Study 2019.

	*Ever Substance Use (Any) ^a^* *(n = 1108)*	*Ever Cigarette Use* *(n = 1108)*	*Ever EVP Use* *(n = 1108)*	*Ever Alcohol Use* *(n = 1108)*	*Ever Marijuana Use* *(n = 1108)*
	OR (95% CI)	OR (95% CI)	OR (95% CI)	OR (95% CI)	OR (95% CI)
** *Major themes ^b^* **					

*Psychological changes*	0.89 (0.59–1.33)	0.97 (0.73–1.30)	0.75 (0.57–1.00) **	0.89 (0.60–1.32)	0.94 (0.69–1.28)
*Physiological changes*	0.92 (0.61–1.37)	0.86 (0.63–1.18)	1.07 (0.80–1.42)	0.83 (0.56–1.22)	1.10 (0.80–1.52)
*Behavioral changes*	0.93 (0.37–2.38)	0.87 (0.51–1.49)	0.61 (0.36–1.04)	1.24 (0.50–3.08)	1.85 (0.95–3.59)
** *Psychological changes ^c^* **			(*n* = 1108)		
*Affect*			0.98 (0.62–1.53)		
*Cognition*			0.63 (0.34–1.15)		
*Need*			1.16 (0.88–1.53)		
*Self-regulation*			0.66 (0.50–0.87) ***		

** *p* < 0.05; *** *p* < 0.001. ^a^ Ever substance use was defined as ever use of any of the following substances: cigarettes, electronic vapor products (EVP), marijuana, or alcohol. ^b^ Multivariable logistic regression models include all three major themes in the same model and control for age group and sex. ^c^ Multivariable logistic regression models for subcodes were conducted separately (e.g., psychological changes model included only subcodes for psychological changes major theme) and control for age group and sex.

## Data Availability

Data and analysis code can be accessed at https://osf.io/q56jg/.

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
