# Peer review of "Exploring Definitions of “Addiction” in Adolescents and Young Adults and Correlation with Substance Use Behaviors"

_ijerph, 2022, doi:10.3390/ijerph19138075_

Round 1

Reviewer 1 Report

This manuscript examined how adolescents and young adults perceived the definition of "addiction" and investigated how those perceptions were associated with substance use behaviors. Overall, this is an interesting topic and the manuscript is well written. I have a few, relatively minor, questions/comments:

- I believe there is a typo in the age grouping for young adults shown on line 94.

- It is somewhat misleading on line 155 to say, "Most participants reported ever use of cigarettes, electronic vapor products....", since young adults were more likely to use substances. "Most participants" implies both adolescent and young adult groups combined.

- On line 170, what is meant by "child categories"?

- The title for Table 2 says correlations, but because results from odds ratios are shown, shouldn't these be logistic regression results?

- There was a significant different by sex for the major theme category of Physiological Changes. However, when looking at ORs for subcategories within Physiological Changes, nothing was significant. What do you think may account for this?

- In section 3.4, if you are reporting ORs then you are not running correlations. You are running univariate logistic regression models with one outcome and one predictor.

- The Table 3 label needs to be corrected as well, since these are univariate logistic regression analyses.

- Given that rates of substance use were much higher in the young adult group, it would be useful for Table 3 analyses to be run separately in young adult and adolescent groups. Even though you controlled for age, because groups were so imbalanced it seems like it would be better to run these analyses separately by age group. 

Author Response

We thank you for your careful and encouraging review of this paper. Please find point-by-point responses to individual comments below. 

  1. I believe there is a typo in the age grouping for young adults shown on line 94.
    • The typo on line 94 regarding the definition of age groups has been corrected.
  2. It is somewhat misleading on line 155 to say, "Most participants reported ever use of cigarettes, electronic vapor products....", since young adults were more likely to use substances. "Most participants" implies both adolescent and young adult groups combined.
    • Regarding substance use rates in our sample, we have retained the total rates of substance use across both age groups, and have also added rates of substance use for youth and young adults separately in the text. We hope that this gives both a sense of the overall composition of our sample as well as the particular use patterns of the separate age groups.
  3. On line 170, what is meant by "child categories"?
    • We have clarified in text the meaning of "child categories," indicating that these are further subordinate categories nested within subcategories (e.g., "dependence: functioning" is subordinate to the "dependence" subcategory, which itself is subordinate to the "physiological changes" category). We hope that this change clarifies the structure of our coding categories.
  4. "The title for Table 2 says correlations, but because results from odds ratios are shown, shouldn't these be logistic regression results? . . .In section 3.4, if you are reporting ORs then you are not running correlations. You are running univariate logistic regression models with one outcome and one predictor. . . The Table 3 label needs to be corrected as well, since these are univariate logistic regression analyses."
    • In all instances where the language of "correlation" was used, we have clarified our language to indicate that we examined the relationships between variables using logistic regression.
  5. There was a significant different by sex for the major theme category of Physiological Changes. However, when looking at ORs for subcategories within Physiological Changes, nothing was significant. What do you think may account for this?
    • Regarding the lack of significant differences by sex in the Physiological Changes subcategory, we note that cell sizes for subcategories were much smaller than for major categories, resulting in non-significant differences between the sexes at the level of subcategory, but which are significant in the aggregate (i.e., at the level of major thematic category).
  6. Given that rates of substance use were much higher in the young adult group, it would be useful for Table 3 analyses to be run separately in young adult and adolescent groups. Even though you controlled for age, because groups were so imbalanced it seems like it would be better to run these analyses separately by age group. 
    • We have run these analyses and have included the results in the text of the paper. 

We thank you again for the time and care taken in this review and hope that the above changes serve to make this a clearer and more robust work.

Reviewer 2 Report

I would like to give congratulations for the authors about this manuscript. I consider that it is a very good job. Even some examples are offered for every kind of definition, I miss a "type definition" in each category that could be use in others "scientific manuscripts". 

Author Response

  1. I would like to give congratulations for the authors about this manuscript. I consider that it is a very good job. Even some examples are offered for every kind of definition, I miss a "type definition" in each category that could be use in others "scientific manuscripts". 
    • Thank you for your kind words and careful review. We refer you to the first column of Table 1, which names the categories/subcategories and also provides definitions for each. We hope that these definitions can be used in future research. 

Reviewer 3 Report

Exploring definitions of “addiction” in adolescents and young adults and correlation with substance use behaviors

The purpose of this research is to perform a comparison on how young people, divided into age groups, describe addiction in their own words. Furthermore, the authors look for a correlation between their definitions and their substance use behaviour.

Comments:

The manuscript is interesting and presents new and useful contexts. There are no publications in the literature on the subject. English language and style are fine.

Only small improvements are recommended as follows:

•          The format of all references should follow the style request of the journal.

•          Insert space between “referrals” and “[31]” in page 2 line 76;

•          Insert space between “software” and “[32]” in page 3 line 112.

Author Response

Thank you for your kind and careful evaluation of this manuscript. Please see point-by-point responses to comments below:

  1.  The format of all references should follow the style request of the journal.
    • Thank you for bringing these discrepancies to our attention; we have made the requested changes.
  2. "Insert space between “referrals” and “[31]” in page 2 line 76; . . . Insert space between “software” and “[32]” in page 3 line 112."
    • We have corrected these typos.